# Fisetin as an Antiviral Agent Targeting the RNA-Dependent RNA Polymerase of SARS-CoV-2: Computational Prediction and In Vitro Experimental Validation

**DOI:** 10.3390/microorganisms13122809

**Published:** 2025-12-10

**Authors:** Ximena Hernández-Rodríguez, Flor Itzel Lira-Hernández, José Manuel Reyes-Ruíz, Juan Fidel Osuna-Ramos, Carlos Noe Farfán-Morales, Daniela Nahomi Calderón-Sandate, Julio Enrique Castañeda-Delgado, Moisés León-Juárez, Rosa María del Ángel, Bruno Rivas-Santiago, Saúl Noriega, David Mauricio Cañedo-Figueroa, Sarita Montaño, Alan Orlando Santos-Mena, Ana Cristina García-Herrera, Luis Adrián De Jesús-González

**Affiliations:** 1Laboratorio de Virología Molecular, Unidad de Investigación Biomédica de Zacatecas, Instituto Mexicano del Seguro Social, Zacatecas 98000, Mexico; 37183498@uaz.edu.mx (X.H.-R.); flor.lihe@gmail.com (F.I.L.-H.); 20202095@uaz.edu.mx (D.N.C.-S.); davidmauricio013@gmail.com (D.M.C.-F.); alan.santos.mena@outlook.com (A.O.S.-M.); ana.garciaher@imss.gob.mx (A.C.G.-H.); 2Unidad Médica de Alta Especialidad, Hospital de Especialidades No. 14, Centro Médico Nacional “Adolfo Ruiz Cortines”, Instituto Mexicano del Seguro Social (IMSS), Veracruz 91897, Mexico; jose.reyesr@imss.gob.mx; 3Facultad de Medicina, Región Veracruz, Universidad Veracruzana (UV), Veracruz 91700, Mexico; 4Faculty of Medicine, Autonomous University of Sinaloa, Culiacán 80246, Mexico; osunajuanfidel.fm@uas.edu.mx; 5Laboratorio de Virología y Diseño de Antivirales, Facultad de Medicina, Universidad Autónoma de Sinaloa (UAS), Culiacán 80246, Mexico; 6Departamento de Ciencias Naturales, Universidad Autónoma Metropolitana (UAM), Unidad Cuajimalpa, Mexico City 05348, Mexico; cfarfan@cua.uam.mx; 7Department of Infectomics and Molecular Pathogenesis, Center for Research and Advanced Studies (CINVESTAV-IPN), Mexico City 07360, Mexico; rmangel@cinvestav.mx; 8Unidad de Investigación Biomédica de Zacatecas, Instituto Mexicano del Seguro Social, Zacatecas 98000, Mexico; julioenriquecastaneda@gmail.com (J.E.C.-D.); rondo_vm@yahoo.com (B.R.-S.); 9Laboratorio de Virología Perinatal y Diseño Molecular de Antígenos y Biomarcadores, Departamento de Inmunobioquimica, Instituto Nacional de Perinatología “Isidro Espinosa de los Reyes”, Mexico City 11000, Mexico; moisesleoninper@gmail.com; 10Unidad Académica de Ciencias Químicas, Universidad Autónoma de Zacatecas, Zacatecas 98160, Mexico; saul.noriega@uaz.edu.mx; 11Laboratory of Molecular Modeling and Bioinformatics, Facultad de Ciencias Químico-Biológicas of Universidad Autónoma de Sinaloa, Culiacán 80246, Mexico; mmontano@uas.edu.mx

**Keywords:** Fisetin, SARS-CoV-2, RNA-dependent RNA polymerase, molecular docking, and antiviral activity

## Abstract

SARS-CoV-2 continues to evolve into immune-evasive variants, and although vaccination remains the cornerstone of prevention, the search for antiviral molecules targeting conserved viral enzymes remains essential. The RNA-dependent RNA polymerase (NSP12) is a central component of coronavirus replication, and natural polyphenols have been recurrently proposed as modulators of viral polymerases. Among these compounds, Fisetin has been reported to interact with multiple viral and cellular pathways, yet its direct antiviral activity against SARS-CoV-2 remained largely unexplored. Here, we first analyzed the interaction of Fisetin with the catalytic and NiRAN domains of NSP12 using molecular docking and molecular dynamics simulations, revealing stable and energetically favorable binding throughout a 100 ns simulation. Previous biochemical reports have shown that Fisetin inhibits the recombinant SARS-CoV-2 RdRp, supporting its potential to engage the polymerase. We then evaluated its antiviral activity in human A549 lung epithelial cells infected with the Omicron JN.1 variant. We observed a clear dose-dependent reduction in viral infection, achieving up to 91.9% inhibition at 3 μM while maintaining acceptable cell viability. In addition, Fisetin displayed a selectivity index superior to that of Lopinavir, the positive antiviral control used in this study. Altogether, our findings demonstrate that Fisetin possesses reproducible antiviral activity in a physiologically relevant human lung model and support its role as a natural scaffold for the rational development of polymerase-targeting antivirals against emerging SARS-CoV-2 variants.

## 1. Introduction

Although coronavirus disease 2019 (COVID-19) no longer represents a pandemic, it continues to cause sporadic outbreaks and local transmission worldwide. According to the World Health Organization (WHO), COVID-19 was declared a public health emergency of international concern in 2020 and has caused more than 7 million deaths globally. Despite the progressive reduction in severe cases due to widespread vaccination and natural immunity, SARS-CoV-2 remains an essential pathogen with ongoing health and economic impacts [1,2,3].

Vaccination constitutes the first line of defense against SARS-CoV-2 infection; however, its protection can wane over time and is often reduced against newly emerging variants of the virus. Consequently, the development of effective antiviral agents is essential as a complementary strategy to prevent severe outcomes, limit viral replication, and reduce transmission, particularly in high-risk or immunocompromised individuals [4,5,6,7].

COVID-19 is caused by *severe acute respiratory syndrome coronavirus 2* (SARS-CoV-2), an enveloped, positive-sense single-stranded RNA virus belonging to the *Coronaviridae* family [8]. The viral genome encodes several structural and non-structural proteins, including the spike glycoprotein (S), the 3C-like protease (NSP5 or Mpro), and the RNA-dependent RNA polymerase (NSP12), which have been widely explored as key therapeutic targets for drug repurposing and rational antiviral design [9,10,11,12,13].

In particular, NSP12 protein is the RNA-dependent RNA polymerase (RdRp or RNA polymerase), responsible for the replication and transcription of genomic RNA [11,14]. This enzyme retains the preserved architecture of viral RNA polymerases. It is composed of the domains fingers, palm, thumb, and NiRAN (Nidovirus RdRp-associated nucleotidyltransferase), which is present only in nidoviruses such as SARS-CoV-2, and interface domains [11,14]. Therefore, it has been described that some antivirals, whose primary function is to inhibit this enzyme, may have a broad spectrum of activity against RNA viruses [15,16,17].

Antiviral drugs can be designed to target either host cell proteins required for viral replication or specific viral proteins. The former approach may offer a broader spectrum of antiviral activity and a lower likelihood of resistance development. However, inhibiting host cell proteins often leads to cytotoxicity and adverse effects on cell viability. Consequently, antivirals directed toward viral proteins are generally more specific and less toxic [18,19,20,21].

Although research activities have fully resumed worldwide, experimental studies involving SARS-CoV-2 still require biosafety level 3 (BSL-3) laboratories for in vitro and in vivo testing, which limits accessibility and throughput [22,23]. In this context, bioinformatics-based strategies have become crucial for identifying and characterizing potential antiviral candidates before experimental validation. Notably, molecular docking and molecular dynamics simulations are among the most powerful computational tools used in modern drug discovery and rational antiviral design [24].

Although several antiviral drugs, such as nirmatrelvir and remdesivir, have been approved for the treatment of COVID-19, their efficacy is limited to specific stages of infection, and resistance or reduced activity against emerging variants has been reported. Furthermore, the availability of effective and affordable antivirals remains a challenge in many regions [20].

Therefore, the repurposing of FDA-approved drugs using bioinformatic and computational pharmacology approaches represents a promising and cost-effective strategy to identify compounds capable of inhibiting SARS-CoV-2 replication [12,13,25].

In recent years, secondary metabolites of plants, such as Fisetin and Hyperoside flavonoids, have demonstrated antiviral effects in vivo and in vitro [17,18] against RNA virus-like dengue virus (DENV) by acting on the NS5 RNA polymerase. Moreover, its antiviral effects have also been observed in other viral infections such as Enterovirus, Rhinovirus, Ebola virus, Chikungunya, and Zika virus [26,27,28,29,30]. Other examples of antiviral compounds are the nucleoside analogs such as Balapiravir, alternative substrates for viral polymerases that competitively inhibit the synthesis of viral RNA in infections such as hepatitis C virus (HCV) and DENV [31,32]. Therefore, both Fisetin and Balapiravir could be excellent candidates against SARS-CoV-2. Moreover, quercetin, a dietary polyphenolic flavonoid found in fruits and vegetables, has been reported to inhibit SARS-CoV-2 infection by interfering with viral entry and replication, including by inhibiting the viral polymerase and proteases [33,34].

Based on the structural and functional conservation of RNA-dependent RNA polymerases among positive-sense RNA viruses, we hypothesized that compounds previously reported as antivirals against other RNA viruses could also inhibit the replication of SARS-CoV-2. Therefore, this study aimed to evaluate the binding affinity of Balapiravir, Fisetin, and Hyperoside to the SARS-CoV-2 RNA-dependent RNA polymerase, NSP12, using molecular docking and molecular dynamics simulations. In addition, the antiviral activity of Fisetin, which exhibited the highest binding affinity in silico, was further evaluated in vitro to validate its potential inhibitory effect on viral replication.

This integrative computational and experimental approach aims to identify FDA-approved or naturally derived compounds that can interfere with viral replication mechanisms, thereby contributing to the rational discovery of new therapeutic options against SARS-CoV-2.

## 2. Materials and Methods

### 2.1. Structure of NSP5 (3C-like Protease) and NSP12 (RNA Polymerase) from SARS-CoV-2

The structure of RNA polymerase (Electron Microscopy, 3.10 Å resolution) from SARS-CoV-2 was obtained from the Protein Data Bank library (RCSB PDB) with PDB ID: 6NUR.

### 2.2. Drug Candidates Tested

Four drug candidates with inhibitory activity against RNA polymerase were selected through literature reviews of drugs with inhibitory activity against viral proteins, specifically those targeting DENV, and the DenvInD database [35,36,37]. These drugs were selected by discarding the drugs already considered against SARS-CoV-2 RNA polymerase in other in silico, in vitro, and in vivo studies. Additionally, non-FDA-approved drugs were also excluded, leaving only Balapiravir, Fisetin, and Hyperoside for analysis against SARS-CoV-2 RNA polymerase.

The selected drugs were obtained from the ZINC15 or PubChem database and are listed below with their ID: Balapiravir (ZINC35328439), Fisetin (ZINC39111), Hyperoside (ZINC3973253), Remdesivir (PubChem CID: 121304016).

### 2.3. Molecular Docking of Drugs Against RNA Polymerase and 3C-like Protease of SARS-CoV-2

The Autodock4 and AutoGrid4 software [38,39,40] were used for the molecular docking analysis of drug candidates against the RNA polymerase from SARS-CoV-2.

Previously, the crystalline structure of RNA polymerase was obtained from the PDB and minimized using PyMOL 2.3.3 software [41] and the text editor Kate 2.2 [42], removing water and other molecules associated with the PDB file [43].

The ligand and target structures were then prepared using AutoDockTools 1.5.6 software, which adds polar atoms and charges [39]. The grid box parameters used were 97.115 Å, 97.809 Å, and 94.063 Å (RNA polymerase), with a grid spacing of 0.375 Å. The genetic algorithm parameters used were: Number of GA Runs = 100, Population size = 150, Maximum Number of Evaluations = medium (10,000,000), with the other values used by default. The best interaction model was chosen based on the lowest calculated ΔG.

### 2.4. Visualization of Molecular Docking Results

The molecular docking results were analyzed using AutoDockTools 1.5.6, PyMOL 2.3.3, and UCSF Chimera 1.14 software [31,33,36] to visualize the optimal target-ligand bond formation with the lowest ΔG. The number and type of non-covalent interactions between the drug and the viral protein were obtained using the PLIP software version 2.3.0 [44].

### 2.5. Molecular Dynamics Simulation and Trajectory Analysis

MD simulation (Molecular dynamics) was performed using NAMD 2.8 software [45] with GPU-CUDA and NVIDIA Tesla C2070/Tesla C2075 video cards. The CHARMM22 force field was employed to create the protein topologies, while the TIP3 model was used for the water molecules [46].

The hydrogen atoms were added using the VMD program in the psfgen software version 2.0 [47]. To neutralize the system, the psfgen software placed 27581 water molecules and eight sodium ions. The system was minimized for 1500 steps, followed by equilibration under constant-temperature, constant-pressure (NPT) conditions for 1 ns, with protein and lipid atoms restrained. A 100 ns long MD simulation was run, considering the RNA polymerase protein and the ligand as soluble, without position restraints, under PBC (periodic boundary conditions), and using an NPT ensemble at 310 K. The MD simulation was performed in the Laboratory of Molecular Modeling and Bioinformatics at laboratory 19 of the Facultad de Ciencias Químico Biológicas of Universidad Autónoma de Sinaloa, and in the Hybrid Cluster Xiuhcoatl (http://clusterhibrido.cinvestav.mx, accessed on 26 August 2023) of the CINVESTAV-IPN, México [48].

The structural analysis of the MD simulation, involving the RNA polymerase protein and the ligand, was performed using Carma software version 2.3 [49]. The root mean square deviation (RMSD) and the radius of gyration (Rg), as well as the different snapshots, were obtained. Molecular graphics were performed in SigmaPlot 12.0. The visualization of 3D structures after the dynamics simulation was performed by VMD Version 1.9.3 [47].

### 2.6. Cell Line and Viral Strain

The human alveolar epithelial cell line A549 was used in this study. The cells were cultured in RPMI-1640 medium supplemented with 8% fetal bovine serum (FBS), 1% ampicillin, and 1% streptomycin, maintained at 37 °C in a humidified atmosphere with 5% CO_2_.

The virus used for the infection assays was SARS-CoV-2, specifically the Omicron variant (B.1.1.529) JN.1. The viral strain was propagated in VERO cells, which are derived from the kidneys of African green monkeys (*Chlorocebus sabaeus*). An infection assay, followed by flow cytometry analysis, was used to determine the viral titer [50].

### 2.7. Biosafety and Regulatory Compliance

All experimental procedures involving SARS-CoV-2 were performed under strict biosafety conditions at the Unidad de Investigación Biomédica de Zacatecas (UIBZ) of the Instituto Mexicano del Seguro Social (IMSS). The UIBZ facilities include biosafety level 2 (BSL-2) laboratories for cell culture and biosafety level 3 (BSL-3) laboratories for SARS-CoV-2 infection assays. Both facilities are equipped to provide maximum protection for personnel, the environment, and the surrounding community.

The study protocol was reviewed and approved by the *Comité Nacional de Investigación Científica (CNIC)* of the Instituto Mexicano del Seguro Social, which integrates the ethics, research, and biosafety committees, under registration number **R-2024-785-073**.

All biosafety procedures were conducted in accordance with the WHO *Laboratory Biosafety Manual* (4th edition), the *Biosafety in Microbiological and Biomedical Laboratories* (BMBL, 6th edition), and the Mexican Official Standards NOM-087-ECOL-SSA1-2002 and NOM-012-SSA3-2012. Infectious and biological waste were managed in accordance with institutional protocols.

### 2.8. Cell Viability Assay

The cellular toxicity of the selected compound, Fisetin, was assessed using the MTT (3-(4,5-dimethylthiazol-2-yl)-2,5-diphenyltetrazole bromide) colorimetric cell viability assay, using the Cell Proliferation Kit I MTT (Roche, REF 11465007001) [51,52].

For the assay, A549 cells were seeded in 96-well plates until they reached 70–80% confluence. Subsequently, they were treated for 24 h at 37 °C with the vehicle (DMSO) and various concentrations of Fisetin (Ambeed, Cat. No: A102354) dissolved in RPMI medium: 1 µM, 1.5 µM, 2 μM, 2.5 μM, 5 μM, 10 μM, and 20 μM. 30% DMSO in RPMI was used as a death control, and fresh RPMI medium as a viability control [50].

After the exposure time, 10 μL of MTT was added to each well and incubated for 4 h at 37 °C, following the manufacturer’s instructions. Subsequently, 100 μL of the solubilization buffer included in the kit was added, and the absorbance was recorded at 620 nm in a microplate spectrophotometer [51,52].

Likewise, a control assay with Lopinavir (Ambeed, Cat. No.: A330804) was included. Although previous clinical assessments have indicated that lopinavir/ritonavir does not confer clinical benefit in COVID-19 patients and that dose adjustment would not improve outcomes due to lopinavir’s very low unbound C_max_, several independent studies have consistently shown that lopinavir exhibits measurable antiviral activity in vitro. Notably, Zhang et al. (2020), and Kang et al. (2020) reported that lopinavir/ritonavir displayed the strongest inhibitory effect among the protease inhibitors tested in SARS-CoV-2–infected Vero E6 and Huh7 cells, supporting its use as a reference positive control in cell-culture antiviral assays [53,54]. Since no cytotoxicity data were available for A549 cells, concentrations of 10 μM, 20 μM, 60 μM, 80 μM, and 100 μM were evaluated under the same experimental conditions described above.

### 2.9. Antiviral Activity Assay

A549 cells were seeded in 24-well plates and, when they reached 70–80% confluence, were infected with SARS-CoV-2, Omicron variant JN.1, at a multiplicity of infection (MOI) of 1 in Hank’s medium for 1 h at 37 °C. The cells were subsequently treated for 24 h at 37 °C with the vehicle (DMSO) and fisetin dissolved in RPMI medium at concentrations of 0.5 μM, 1 μM, 1.5 μM, 2 μM, 2.5 μM, 3 μM, and 3.5 μM [53].

After treatment, the cells were harvested by centrifugation, the medium was removed, and the cells were washed with 1X PBS. They were then fixed with 4% paraformaldehyde for 30 min at room temperature. The cells were then permeabilized with a 0.02% saponin solution containing 1% FBS in 1X PBS for 30 min, followed by incubation with the primary anti-Spike monoclonal antibody (1:750, GeneTex, Cat. No: GTX632604) for 24 h at 4 °C [55].

The samples were then washed with 1X PBS and incubated with the Alexa Fluor 647-conjugated secondary antibody (1:200, Abcam, Cat. No: AB150115) for 1 h at room temperature. Finally, the cells were rewashed with 1X PBS, resuspended in 200 μL of 1X PBS, and analyzed by flow cytometry. The control assay was performed with Lopinavir under the same experimental conditions, at concentrations of 3 μM, 6 μM, 9 μM, 12 μM, 15 μM, and 18 μM [51].

### 2.10. Statistical Analysis

For the in silico assays, molecular docking was performed in six independent runs per compound, generating six binding free energy (ΔG) values and six RMSD values for each ligand–protein interaction. The results were summarized as mean ΔG ± standard deviation, and only poses with RMSD < 1 Å were included in the statistical analysis. Differences in mean affinity values among the reference drugs (Remdesivir and Baicalein) and the candidate compounds (Fisetin, Quercetin, Hyperoside, Balapiravir, and Anthraquinone) were assessed using a one-way ANOVA, followed by Dunnett’s multiple comparison test, with Remdesivir as the reference control. Statistical analyses were performed using GraphPad Prism version 9, and significance was defined as *p* < 0.05 [21]. For the in vitro assays, data obtained from the antiviral activity and cell viability experiments were collected and analyzed using a BD FACSCanto II flow cytometer (Clinical Flow Cytometry System, BD Biosciences, New York, US) and processed with FACSdiva software v9.0. Statistical analyses were performed with R Studio version 4.3.1 and GraphPad Prism version 9.5.1 [51].

The CC_50_ (50% cytotoxic concentration) and IC_50_ (50% inhibitory concentration) values were calculated by nonlinear regression analysis using a sigmoidal dose–response model. To determine statistically significant differences between treatments and controls, a one-way ANOVA was conducted, followed by Tukey’s post hoc test. All data were normalized to the untreated control, and results were expressed as the relative percentage of infection, with 100% representing the infection level in untreated cells [51].

## 3. Results

Given the critical functions of SARS-CoV-2 viral proteins, such as the type 3C protease (Mpro) and RNA-dependent RNA polymerase (RdRp or NSP12), these proteins have been widely proposed as drug targets [9,10]. NSP12 is responsible for the replication and transcription of the viral genome and, together with its cofactors NSP7 and NSP8, forms the replicase complex necessary to maintain polymerase activity [14].

Considering that RNA-dependent RNA polymerase (RdRp) is a highly conserved enzyme among different positive-strand RNA viruses, the possibility of identifying potential inhibitors of SARS-CoV-2 RdRp from compounds with previously described antiviral activity against dengue virus (DENV) was explored [56,57].

For this purpose, we used the DenvInD database, which includes 528 drugs with experimental evidence of antiviral activity against DENV nonstructural proteins. This database is handy for repurposing studies, as many antivirals active against flaviviruses share mechanisms of action that target RNA replication enzymes [37].

Using molecular docking techniques, we analyzed the affinity between SARS-CoV-2 NSP12 (PDB ID: 6NUR). We selected drugs from the DenvInD database, including Balapiravir, Fisetin, and Hyperoside, all of which have been reported to have antiviral activity against DENV [35,36].

From an initial set of 160 compounds with inhibitory potential, those already evaluated against SARS-CoV-2 in previous studies (in silico, in vitro, or in vivo) were excluded, as well as compounds not approved by the FDA. Finally, Balapiravir, Fisetin, and Hyperoside were selected for analysis against SARS-CoV-2 RNA polymerase.

### 3.1. Screening of DENV NS5 Polymerase Inhibitors Against SARS-CoV-2 RNA Polymerase

Remdesivir was used as a control drug (in in silico assays) given its described RNA polymerase inhibitory activity and current therapeutic use against SARS-CoV-2 [58,59,60]. The comparison of the affinity of different drugs, analyzed as possible inhibitors of SARS-CoV-2 RNA polymerase, is shown in Appendix A, which includes their PubChem access number, binding energy, toxicity reported by PubChem, and FDA legislation. Additionally, the PLIP software was used to predict possible non-covalent interactions (Appendix A) [44].

Fisetin and Hyperoside are compounds derived from plants and fungi, whereas Balapiravir is a nucleoside 5′-triphosphate analog that competitively inhibits viral RNA synthesis [61,62,63].

All tested drugs, Fisetin, Hyperoside, Balapiravir, and Remdesivir showed affinity for two RNA polymerase sites: the active site and the NiRAN-Finger site (Figure 1).

Fisetin and Balapiravir showed higher affinity for the active site (−7.08 and −7.25 kcal/mol, respectively) (Figure 2C and Figure 2B) compared with Remdesivir (control) (−6.81 kcal/mol) (Figure 2A). Hyperoside showed the lowest affinity (−6.68 kcal/mol) (Figure 2D), although PLIP analysis predicted a high number of non-covalent interactions (Appendix A).

In addition to the high affinity of the tested drugs for the active site, a higher affinity for the NiRAN subdomain was also observed. The NiRAN domain, present only in Nidoviruses, has nucleotidylation activity essential for viral replication; mutations in this domain result in non-infective viral particles [64,65]. Fisetin and Balapiravir again exhibited higher affinity (−8.1 and −8.06 kcal/mol) for the NiRAN-Fingers site (Figure 2G and Figure 2H) compared to Remdesivir (−7.67 kcal/mol) (Figure 2F).

The affinity of the drugs for both sites was plotted (Figure 3A,B). Fisetin showed the highest affinity for the active site (−7.042 ± 0.035 kcal/mol) and the NiRAN domain (−8.1 kcal/mol).

### 3.2. Analysis of Molecular Dynamics of Fisetin Bound to SARS-CoV-2 RNA Polymerase

Given that Fisetin displayed the highest affinity for both RNA polymerase functional sites (Figure 2C–F and Figure 3A,B), and formed the most significant number of predicted non-covalent interactions (ST2), we further evaluated the stability of its binding to the NiRAN domain (−8.1 kcal/mol) through molecular dynamics (MD) simulations.

From the three prioritized compounds (Balapiravir, Fisetin, and Hyperoside), Fisetin was selected for cellular evaluation due to its more favorable pharmacological profile. Although Balapiravir showed comparable binding energies, its clinical development was discontinued in hepatitis C and dengue due to significant hematological toxicity, including lymphopenia, limiting its potential for antiviral repurposing [32,66,67]. Meanwhile, Hyperoside, a more polar glycosylated flavonoid, exhibits reduced membrane permeability and lower intracellular accumulation compared with its aglycone forms, thereby reducing the likelihood of achieving effective intracellular concentrations at the RdRp site [68,69,70]. In contrast, Fisetin combines high predicted binding stability and feasible intracellular accessibility; therefore, we proceeded with MD simulations, cytotoxicity assays, and antiviral evaluation in A549 cells.

The 100 ns MD trajectory showed that the RMSD of the RNA polymerase–Fisetin complex reached equilibrium within the first nanosecond (Figure 4A). The Rg profile indicated expansion between 30 and 50 ns and compaction from 55 to 70 ns, with the system remaining stable thereafter (Figure 4B). Snapshots at 20, 40, 60, 80, and 100 ns confirmed that Fisetin remained in the binding site throughout the simulation (Figure 4C).

### 3.3. Fisetin Cytotoxicity Profile in A549 Lung Cells

Following the in silico analyses, we conducted in vitro assays to evaluate the antiviral potential of Fisetin against SARS-CoV-2. As an initial step, the cytotoxicity of Fisetin was assessed in the human lung epithelial cell line A549 to determine concentrations that did not significantly compromise cell viability. Cells were exposed to a concentration range between 1 μM and 20 μM.

Fisetin exhibited low cytotoxicity within the range of 1 μM to 2 μM, maintaining cell viability above 80% (Figure 5B). Notably, only the highest tested concentration (20 μM) resulted in a viability below 50%. The median cytotoxic concentration (CC_50_) was determined through nonlinear regression analysis using normalized absorbance values, yielding a CC_50_ of 12.10 μM in A549 cells (Figure 5D).

As a control, a parallel assay was performed with Lopinavir (a positive control with anti-SARS-CoV-2 activity [53]), which exhibited a CC_50_ of 82.72 μM, indicating higher cytotoxicity compared to Fisetin (Figure 5A,C).

Based on these results, concentrations maintaining >80% viability (1 μM, 1.5 μM, 2 μM, 2.5 μM, and 3 μM) were selected for testing antiviral activity.

### 3.4. Fisetin Exhibits Antiviral Activity Against SARS-CoV-2 in A549 Lung Cells

The antiviral effect of Fisetin against SARS-CoV-2 was evaluated in A549 cells infected with the Omicron JN.1 variant, using the six low cytotoxic concentrations identified in the viability assay. ANOVA and Tukey’s post hoc test showed that all concentrations significantly reduced viral infection compared to the untreated control. A clear dose-dependent trend was observed, with statistically significant differences among concentrations.

The percentage reduction in infection ranged from 24.42% at 1 μM to 91.95% at 3 μM. Based on these findings, 2 μM was identified as the optimal concentration, maintaining safe viability levels while achieving a 71.76% reduction in infection (Figure 6B), outperforming lopinavir (an antiviral with reported anti-SARS-CoV-2 activity) (Figure 6A).

Median inhibitory concentrations (IC_50_) were calculated to determine the selectivity index (SI) for both compounds. Lopinavir exhibited an SI of 4.86, whereas Fisetin achieved an SI of 8.039, indicating a broader therapeutic window and enabling the exploration of a wider concentration range without compromising cell viability (Figure 5).

Importantly, IC_50_ values for both Fisetin and Lopinavir were below their respective CC_50_ values, confirming that antiviral activity occurs at concentrations well below those causing cytotoxicity.

These results highlight Fisetin as a selective antiviral candidate with a favorable in vitro balance between antiviral potency and cytotoxicity, and with a higher selectivity index than Lopinavir under identical conditions

## 4. Discussion

The global impact of COVID-19 has transformed our understanding of viral pathogenesis and therapeutic development. Although the pandemic phase has officially ended, SARS-CoV-2 remains endemic in several regions, continuing to cause sporadic outbreaks and hospitalizations [71]. Vaccination remains the cornerstone of prevention, yet antiviral agents continue to be essential, particularly for immunocompromised individuals and populations with incomplete vaccine coverage [72]. This dual strategy ensures not only the reduction in viral transmission but also mitigates disease progression in acute infections.

The identification of viral enzymes as molecular targets has guided most antiviral strategies. Among these, RNA-dependent RNA polymerase (RdRp or NSP12) is one of the most conserved and essential enzymes for SARS-CoV-2 replication, catalyzing the synthesis of genomic and subgenomic RNAs [65]. This conservation among positive-strand RNA viruses, including flaviviruses, makes RdRp an attractive target for drug repositioning. Accordingly, we explored potential inhibitors of SARS-CoV-2 RdRp derived from compounds previously characterized as active against dengue virus (DENV), using the DenvInD database as a source of candidate molecules [37].

Our molecular docking results revealed that Fisetin and Balapiravir displayed higher binding affinities for both the active and NiRAN domains of SARS-CoV-2 RNA polymerase compared to remdesivir, a clinically approved RdRp inhibitor. The NiRAN domain, unique to nidoviruses, plays a crucial role in RNA capping and nucleotide transfer reactions, and its inhibition has been linked to a loss of viral infectivity [64]. Fisetin, a flavonol of plant origin, exhibited the strongest affinity, with interaction energies of approximately −7.4 and −8.1 kcal/mol for the catalytic and NiRAN domains, respectively, accompanied by multiple noncovalent bonds that stabilized the complex.

These findings are consistent with previous in silico reports describing the high binding capacity of Fisetin to the polymerase active site in other RNA viruses, including DENV and ZIKV [73,74,75]. The stability of these interactions, further confirmed by our 100 ns molecular dynamics simulation, supports the possibility that Fisetin interferes with the polymerase’s catalytic mechanism by maintaining persistent contact with critical residues within the NiRAN-Fingers interface. This observation aligns with the structural behavior of other flavonols, such as quercetin, which also display stable binding to RdRp motifs A through G [76].

Balapiravir, a nucleoside analog originally designed for HCV therapy, also demonstrated high affinity for the same polymerase domains. However, its clinical efficacy against DENV was limited, possibly due to inefficient metabolic activation in host cells [66]. This observation suggests that while Balapiravir has potential inhibitory capacity at the molecular level, its practical application against SARS-CoV-2 may depend on pharmacokinetic factors. In contrast, Fisetin’s non-nucleoside structure may confer better cellular accessibility and a more favorable safety profile for potential therapeutic use.

Collectively, these computational findings underscore the feasibility of identifying RdRp inhibitors through cross-viral databases, leveraging the structural conservation between RNA polymerases. They also provide a strong theoretical foundation for validating the antiviral potential of Fisetin through biological assays, bridging virtual screening with experimental virology.

The integration of in silico and in vitro analyses provides a comprehensive framework for identifying antiviral agents with real translational potential. Following the computational predictions, our experimental assays confirmed that Fisetin effectively reduced SARS-CoV-2 replication in A549 lung epithelial cells infected with the Omicron JN.1 variant. The inhibition showed a clear dose-dependent response, with reductions of 71.7% at 2 µM and 91.9% at 3 µM, while maintaining more than 80% cellular viability. These findings demonstrate not only the biological relevance of the in silico predictions but also the capacity of Fisetin to act as a selective and safe antiviral compound.

The calculated selectivity index for Fisetin was 8.039, higher than that of Lopinavir, which reached 4.86 under identical conditions. This result indicates a more favorable efficacy-to-toxicity ratio, suggesting that Fisetin exerts its antiviral effect at concentrations well below those that cause cytotoxicity. Similar indices have been reported for other plant-derived flavonols, such as quercetin and luteolin, which display in vitro antiviral effects in the low micromolar range [71,76]. These data support the concept that natural compounds with polyphenolic structures can serve as efficient inhibitors of viral replication while preserving cell viability.

The observed antiviral activity may derive primarily from Fisetin’s interaction with RNA polymerase. Docking and molecular dynamics results suggest that Fisetin stabilizes within the NiRAN domain and the catalytic cleft of NSP12, potentially interfering with nucleotide binding or elongation during RNA synthesis. Such interactions could hinder the formation of replication complexes, resulting in decreased RNA viral levels. This mechanism is consistent with reports that describe how inhibition of the NiRAN site disrupts viral transcription and replication in nidoviruses [64,65].

Importantly, our findings are consistent with previously reported biochemical evidence demonstrating that Fisetin directly inhibits the SARS-CoV-2 RdRp. Under its commercial formulation as *Fustel*, Fisetin exhibited ~95.8% inhibition of recombinant RdRp at 20 μM and an IC_50_ of 2.17 μM, while also displaying partial inhibition of the viral helicase (39.3% at 20 μM) [72]. This prior enzymatic validation provides independent mechanistic support for the polymerase-targeting activity suggested by our molecular docking and MD simulations. Our study extends these biochemical observations by confirming that Fisetin exerts antiviral activity in human lung epithelial cells infected with the Omicron JN.1 variant, thereby bridging enzymatic inhibition with physiologically relevant intracellular antiviral efficacy.

Several structurally related flavonol aglycones have also been explored as anti-SARS-CoV-2 agents. Quercetin has been reported to interact with RdRp and to modulate entry-associated pathways; myricetin directly inhibits the NSP13 helicase, as demonstrated by crystallographic studies; and kaempferol has shown activity against viral entry/fusion and moderate biochemical inhibition of Mpro. Together, these aglycone flavonols illustrate a multi-target antiviral potential within this chemical family [34,77,78,79,80].

Glycosylated homologues such as Hyperoside and Rutin have been proposed as inhibitors of Mpro and Spike/ACE2 interfaces in computational and biochemical studies, and some evidence suggests interference with host entry proteases. However, their increased polarity reduces membrane permeability and intracellular accumulation, making them less likely to engage intracellular targets such as RdRp or NSP13 than aglycones like Fisetin [68,81,82,83,84].

In addition to its polymerase-directed effects, the antiviral activity observed in our assays may also reflect complementary host-response modulation, a biological property that has been extensively described for Fisetin in non-viral contexts. Previous studies have shown that Fisetin can influence key signaling pathways involved in inflammation and cellular homeostasis, including PI3K/Akt, NF-κB, and Nrf2, all of which are known to be exploited or dysregulated during coronavirus infection [85]. However, these pathways were not directly evaluated in the present study, and their potential contribution to the anti-SARS-CoV-2 activity observed here remains hypothetical. Future work will incorporate targeted analyses of gene and protein expression to determine whether Fisetin modulates these signaling cascades during infection, and to dissect the relative contribution of viral and host-directed mechanisms to its antiviral profile. Our results are comparable to those of other reported studies, which demonstrated that natural flavonoids maintain stable interactions with the catalytic motifs of SARS-CoV-2 RdRp and exhibit significant viral inhibition in lung-derived cell lines. These structural characteristics are present in Fisetin, supporting the consistency between molecular modeling and biological observations [86].

The use of the A549 cell line, derived from human alveolar epithelium, provides translational relevance by representing the primary anatomical site of SARS-CoV-2 infection. Although A549 cells display lower ACE2 and TMPRSS2 expression than Vero or Calu-3 cells, multiple transcriptomic and functional studies have demonstrated that SARS-CoV-2 can infect A549 through ACE2-independent pathways, including entry routes mediated by cathepsins, Neuropilin-1 (NRP1), and C-type lectins (DC-SIGN/L-SIGN) [87,88]. Moreover, infection and replication kinetics of SARS-CoV-2 have been documented in A549 cells across several studies, confirming that this model supports productive viral replication despite reduced canonical receptor expression [53,89]. Their predictable growth, human origin, and compatibility with endosomal entry mechanisms make A549 a suitable platform for mechanistic antiviral evaluation. Consistent with this, other authors have successfully employed A549 cells to test natural antiviral compounds against SARS-CoV-2, reporting reproducible infection and inhibition profiles [71,90].

Taken together, these findings suggest that Fisetin inhibits SARS-CoV-2 replication through a combination of polymerase-directed inhibition and host-response modulation. The convergence of both mechanisms could explain the broad antiviral capacity of this compound, as well as its favorable selectivity profile when compared with standard antivirals.

## 5. Conclusions

This study identifies Fisetin as a polymerase-targeting antiviral candidate scaffold with demonstrated in vitro activity against SARS-CoV-2. Computational analyses showed that Fisetin forms stable interactions with both the NiRAN and catalytic domains of NSP12, and previous biochemical studies have reported micromolar inhibition of recombinant RdRp. Our data extend these mechanistic observations by confirming that Fisetin reduces viral replication in human lung epithelial cells infected with the Omicron JN.1 variant at concentrations that preserve acceptable cell viability.

Rather than proposing Fisetin as a therapeutic product, our findings position it as a biologically validated starting point for the rational development of polymerase-directed antivirals. Future studies should focus on improving its pharmacokinetic properties, defining its precise intracellular mode of action, and assessing its antiviral potential in additional cellular models and in vivo systems.

Collectively, this work supports the advancement of Fisetin as a natural antiviral scaffold, integrating enzymatic evidence, molecular modeling, and cellular antiviral activity, and contributing to ongoing efforts toward the development of broad-spectrum agents against emerging RNA viruses.

## Figures and Tables

**Figure 1 microorganisms-13-02809-f001:**
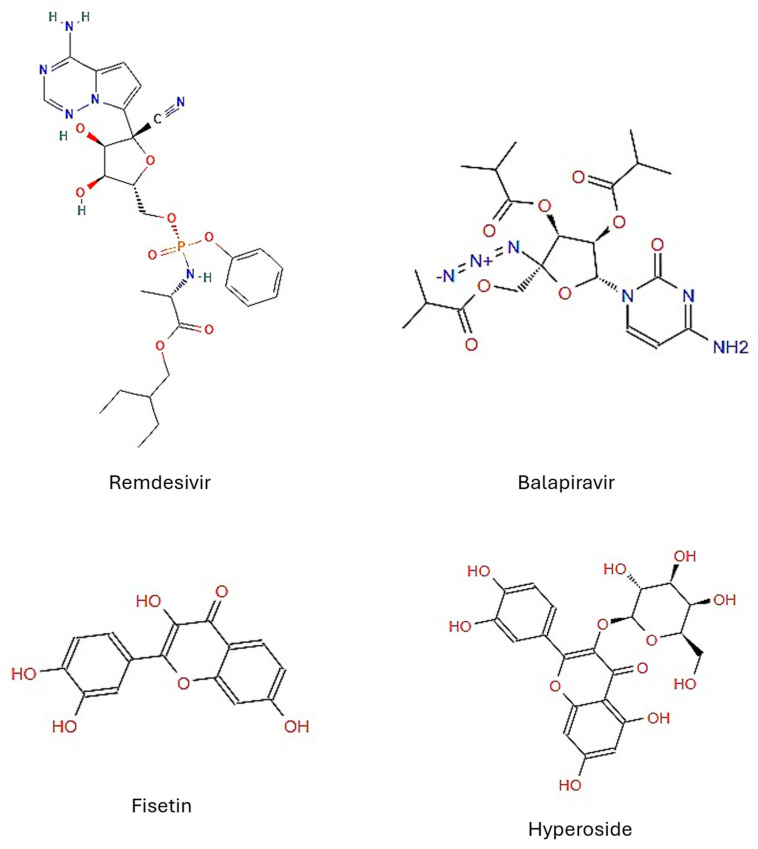
Chemical structures of the drug candidates tested. Structures obtained from ZINC15.

**Figure 2 microorganisms-13-02809-f002:**
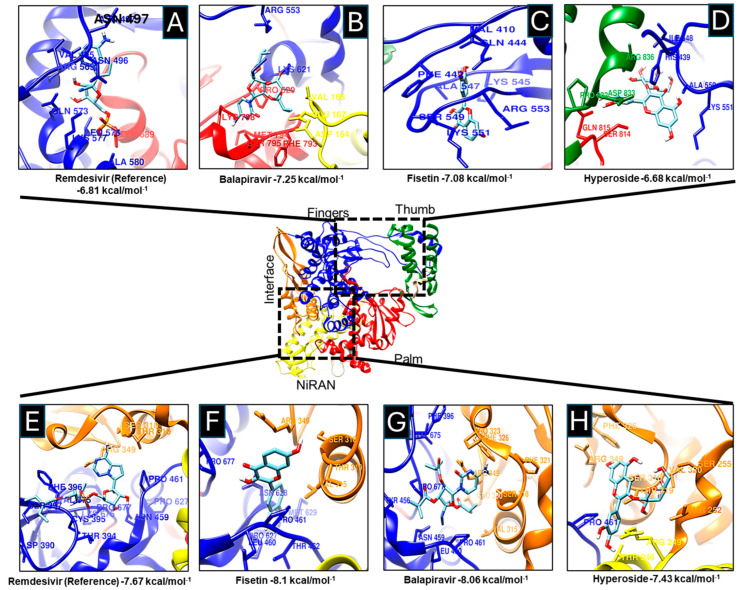
Molecular docking between drugs and SARS-CoV-2 RNA polymerase. The components Remdesivir (**A**,**E**), Balapiravir (**B**,**G**), Fisetin (**C**,**F**), and Hyperoside (**D**,**H**) bind to the active site and the NiRAN domain of the SARS-CoV-2 RNA polymerase. RNA Polymerase Fingers domain (blue), Thumb (green), Palm (red), Interface (orange), and NiRAN (yellow) are shown. Additionally, amino acids have been shown to interact with various drugs and exhibit different binding energies.

**Figure 3 microorganisms-13-02809-f003:**
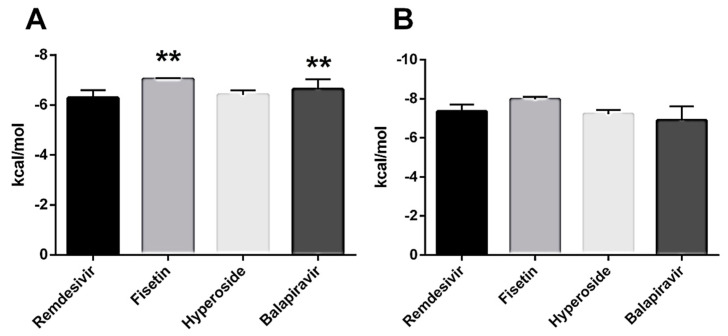
Comparison of Drug Affinities by SARS-CoV-2 RNA Polymerase. (**A**) Drug affinities for the active site of RNA polymerase. (**B**) Affinities of drugs by the NiRAN site. ** *p* < 0.001.

**Figure 4 microorganisms-13-02809-f004:**
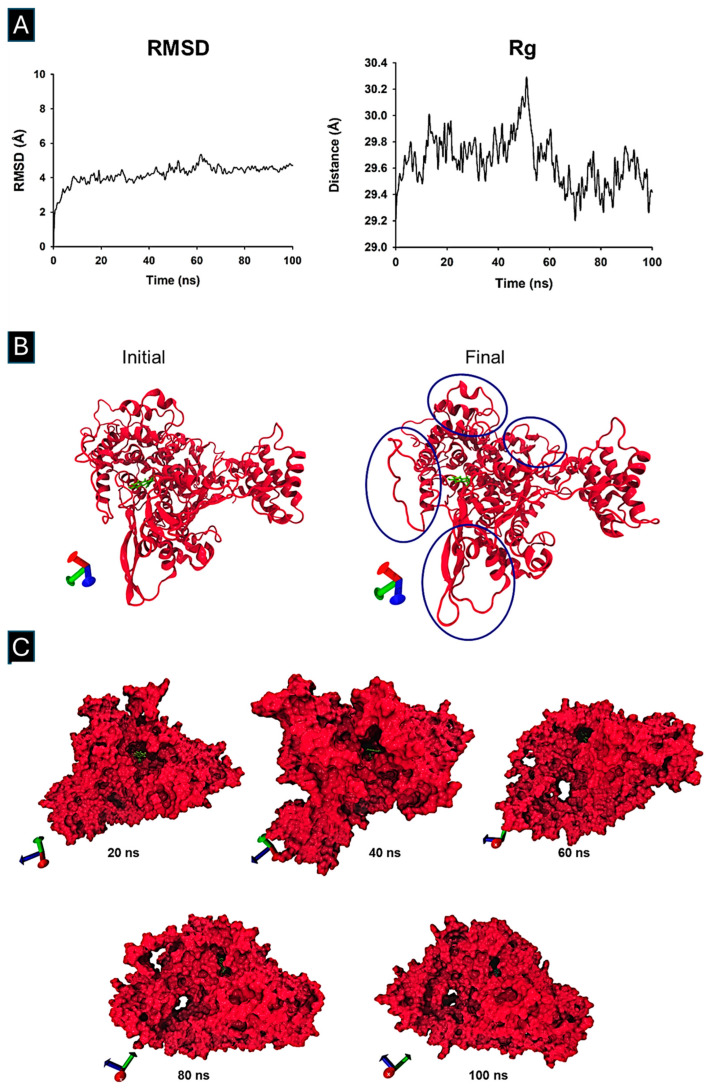
Molecular Dynamic Simulation and Trajectory analysis of Fisetin bound to SARS-CoV-2 RNA Polymerase. The structural analysis of the MD simulation involving the RNA polymerase protein and the ligand was performed using Carma software version 2.3. (**A**) The root mean square deviation (RMSD) and (**B**) the radius of gyration (Rg), marked in blue circles, indicate an expansion of the system, (**C**) as well as the different snapshots at 20, 40, 60, 80, and 100 ns, were obtained. Molecular graphics were performed in SigmaPlot 12.0. The visualization of 3D structures after dynamic simulation was performed using VMD software version 1.9.3.

**Figure 5 microorganisms-13-02809-f005:**
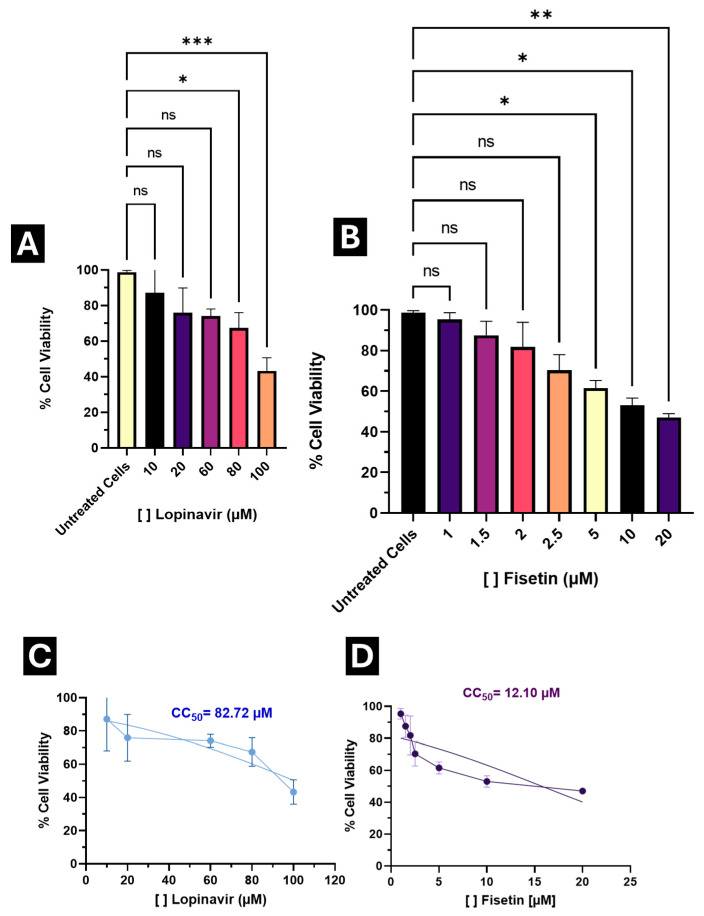
Cytotoxic effect of Lopinavir and Fisetin in A549 cells. (**A**) Percentage of cell viability at different concentrations of Lopinavir; (**B**) Percentage of cell viability to varying concentrations of Fisetin; (**C**) Calculated median cytotoxic concentration (CC_50_) for Lopinavir; (**D**) Calculated median cytotoxic concentration (CC_50_) for Fisetin. Results are expressed as the percentage of cell viability, normalized to the control (Untreated cells). * *p* < 0.05, ** *p* < 0.001, *** *p* < 0.0001, ns: no statistical difference.

**Figure 6 microorganisms-13-02809-f006:**
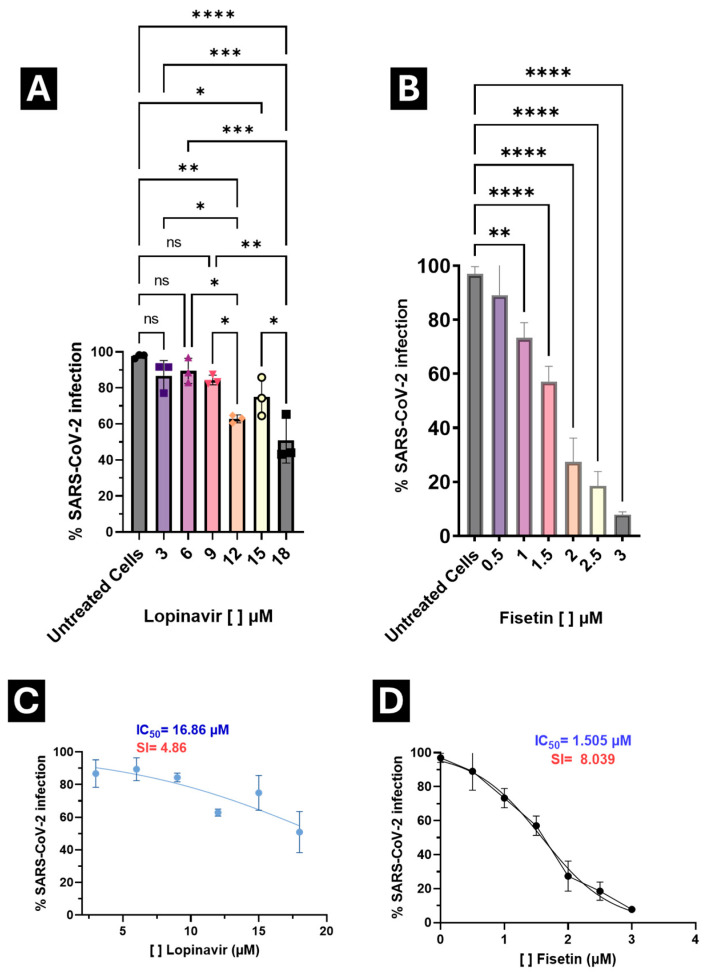
Antiviral activity of Lopinavir and Fisetin against SARS-CoV-2 infection in A549 cells. (**A**) Treatment with Lopinavir at different concentrations; (**B**) Treatment with Fisetin at different concentrations; (**C**) Calculation of the selectivity index (SI) for Lopinavir; (**D**) Calculation of the SI for Fisetin. Viral titers were determined by flow cytometry after infection assays, and results are expressed as the percentage of infected cells, normalized to the control. * *p* < 0.05, ** *p* < 0.001, *** *p* < 0.0001, **** *p* = 0.00001, no statistical difference. The selectivity index (SI) of the drugs was calculated as the CC_50_/IC_50_ ratio.

## Data Availability

The original contributions presented in this study are included in the article/Appendix A. Further inquiries can be directed to the corresponding author.

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
