# Peer review of "Fisetin as an Antiviral Agent Targeting the RNA-Dependent RNA Polymerase of SARS-CoV-2: Computational Prediction and In Vitro Experimental Validation"

_microorganisms, 2025, doi:10.3390/microorganisms13122809_

Round 1
Reviewer 1 Report
Comments and Suggestions for Authors
Luis Adrián De Jesús-González and coworkers reported an initial computational screening of 528 compounds from the DenvInD database, which were identified as potential antivirals targeting the NSP12 protein of SARS-CoV-2. This enzyme represents a suitable target for drug development, as it is highly conserved among viral strains. From this screening, the authors identified Balapiravir, Fisetin, and Hyperoside as promising candidates. Among these, Fisetin emerged as the most interesting molecule, exhibiting the best performance both in computational analyses and experimental assays.
The manuscript is well written and scientifically sound. It aligns well with the journal’s scope and contributes to expanding the structure–activity relationship (SAR) knowledge of the flavonoid class of compounds with anti-SARS-CoV-2 activity. Therefore, I recommend publication after the following minor issues are addressed:
-
Include chemical structures of the tested compounds as graphical representations.
-
Page 2, line 78: Please cite the following reference: 10.2147/DDDT.S450499; 10.1038/s41392-022-01249-8; 10.1038/s41598-024-75519-6;
-
Reference 15 appears off-topic and should be reconsidered.
-
Introduction: Fisetin, a close analogue of quercetin (already known to inhibit SARS-CoV-2 replication), should be briefly discussed and properly referenced. Suggested references include:
-
10.3390/ijms22137048
-
10.3390/molecules26196062
-
10.1016/j.ejmcr.2023.100125
-
10.1371/journal.pone.0312866
-
10.1016/j.ijbiomac.2020.07.235
-
Author Response
Reviewer 1
Luis Adrián De Jesús-González and coworkers reported an initial computational screening of 528 compounds from the DenvInD database, which were identified as potential antivirals targeting the NSP12 protein of SARS-CoV-2. This enzyme represents a suitable target for drug development, as it is highly conserved among viral strains. From this screening, the authors identified Balapiravir, Fisetin, and Hyperoside as promising candidates. Among these, Fisetin emerged as the most interesting molecule, exhibiting the best performance both in computational analyses and experimental assays.
The manuscript is well written and scientifically sound. It aligns well with the journal’s scope and contributes to expanding the structure–activity relationship (SAR) knowledge of the flavonoid class of compounds with anti-SARS-CoV-2 activity. Therefore, I recommend publication after the following minor issues are addressed:
- Include chemical structures of the tested compounds as graphical representations.
Reply: Thank you for your comment. We have added Figure 1, which shows the chemical structures of the drug candidates tested.
- Page 2, line 78: Please cite the following reference: 10.2147/DDDT.S450499; 10.1038/s41392-022-01249-8; 10.1038/s41598-024-75519-6;
Reply: The references suggested were included in this new submission.
- Reference 15 appears off-topic and should be reconsidered.
Reply: Reference #15 was removed, and references 10.1038/s41392-022-01249-8 and 10.2147/DDDT.S450499 were added.
- Introduction: Fisetin, a close analogue of quercetin (already known to inhibit SARS-CoV-2 replication), should be briefly discussed and properly referenced. Suggested references include:
- 10.3390/ijms22137048
- 10.3390/molecules26196062
- 10.1016/j.ejmcr.2023.100125
- 10.1371/journal.pone.0312866
- 10.1016/j.ijbiomac.2020.07.235
Reply: These papers were included in this new submission, as follows:
Moreover, quercetin, a dietary polyphenolic flavonoid found in fruits and vegetables, has been reported to inhibit SARS-CoV-2 infection by interfering with viral entry and replication, including by inhibiting the viral polymerase and proteases (10.1016/j.ejmcr.2023.100125 and 10.1371/journal.pone.0312866).
Reviewer 2 Report
Comments and Suggestions for Authors
In the manuscript ‘Fisetin as an antiviral agent targeting the RNA-dependent RNA polymerase of SARS-CoV-2: computational prediction and in vitro experimental validation’, the authors aim to identify potential inhibitors of the SARS-CoV-2 RNA-dependent RNA polymerase by repurposing compounds previously reported to act against RNA viruses. Based on molecular docking and dynamics simulations, Fisetin resulted the best candidate for subsequent investigations and in vitro experiments revealed a possible antiviral activity.
Main comments:
It is not clear why the authors use Lopinavir as positive control with anti-SARS-CoV-2 activity since, as reported in ref [46] ‘..our research suggested that administration of LPV/r under standard dose could hardly provide any benefit to COVID-19 patients, and adjustment of dosing regimen will not lead to clinical benefit due to the very low unbound Cmax of lopinavir. Further clinical trial of LPV/r is not recommended.’
The authors choose A549 as the only cellular model even it is well reported that express low ACE2 and TMPRSS2 levels. Did the authors investigate other cell lines more suitable for infection assays? Moreover, the Fisetin cytotoxicity profile in A549 lung cell reports that the molecule affects the cell viability starting from the lowest concentrations tested. It is not clear what are the non-cytotoxic concentrations of Fisetin.
At line 377 the authors declare that ‘These results highlight Fisetin as a safe and effective therapeutic alternative against SARS-CoV-2, with a superior selectivity index compared to Lopinavir, and a favorable balance between antiviral potency and cytotoxicity’. Nevertheless, the presented experiments reveal that there is no an effective Fisetin concentration without cytotoxic effects, since 0.5 µM showed no cytotoxic effect but resulted not statistically significant in inhibiting the viral infection. How the authors comment this evidence?
At line 468 the authors conclude that ‘these findings suggest that Fisetin inhibits SARS-CoV-2 replication through a combination of polymerase-directed inhibition and host-response modulation’, possible via PI3K/Ak, NF-ĸB and Nrf2 but they did not check a modulation of these factors via gene or protein expression. How the authors comment this advice?
- It is not clear how the authors calculate statistics in Figure 2
- In Figure 4, the statistic and the Untreated cells are not shown
- Could the authors check if reference [65] is appropriated for Fisetin?
line 298: Remdesivir is shown in Figure1A-1F
line 305 Balapiravir is shown in Figure 1H
line 306: The affinity of the drugs was plotted in Figure 2 A and B)
line 349: The authors mention 0.5 µM Fisetin concentration, but it is not reported in the graph 4B
line 368: Figure 19B is not correct
line 369: It is not clear what is the positive control referred to
line 206: Please correct 1 nM and 1.5 nM concentrations in µM
Author Response
Reviewer 2
In the manuscript ‘Fisetin as an antiviral agent targeting the RNA-dependent RNA polymerase of SARS-CoV-2: computational prediction and in vitro experimental validation’, the authors aim to identify potential inhibitors of the SARS-CoV-2 RNA-dependent RNA polymerase by repurposing compounds previously reported to act against RNA viruses. Based on molecular docking and dynamics simulations, Fisetin resulted the best candidate for subsequent investigations and in vitro experiments revealed a possible antiviral activity.
Main comments:
It is not clear why the authors use Lopinavir as positive control with anti-SARS-CoV-2 activity since, as reported in ref [46] ‘..our research suggested that administration of LPV/r under standard dose could hardly provide any benefit to COVID-19 patients, and adjustment of dosing regimen will not lead to clinical benefit due to the very low unbound Cmax of lopinavir. Further clinical trial of LPV/r is not recommended.’
Reply: We thank the reviewer for this helpful observation. We agree that lopinavir/ritonavir has shown no clinical benefit in COVID-19 therapeutic trials, as noted in reference [46,47]. To avoid any confusion between clinical efficacy and experimental utility, we have clarified in the Materials and Methods section that lopinavir was included solely as an in vitro positive control (Lines 285-295).
To address the reviewer’s concern, we have now explicitly indicated in the revised Methodology that lopinavir was used as an in vitro reference inhibitor, and not as a model of therapeutic performance.
The authors choose A549 as the only cellular model even it is well reported that express low ACE2 and TMPRSS2 levels. Did the authors investigate other cell lines more suitable for infection assays? Moreover, the Fisetin cytotoxicity profile in A549 lung cell reports that the molecule affects the cell viability starting from the lowest concentrations tested. It is not clear what the non-cytotoxic concentrations of Fisetin are.
Reply: We thank the reviewer for this important comment. To clarify this point, we have expanded the Discussion to better justify the rationale for using A549 cells. Although A549 cells express lower levels of ACE2 and TMPRSS2 than Vero or Calu-3 cells, multiple transcriptomic and functional studies have shown that SARS-CoV-2 can infect A549 through ACE2-independent pathways, including entry routes mediated by cathepsins, Neuropilin-1 (NRP1), and C-type lectins (DC-SIGN/L-SIGN). Furthermore, productive SARS-CoV-2 infection and replication kinetics in A549 cells have been documented in several independent studies. Because A549 cells are derived from human alveolar epithelium, the primary anatomical target of SARS-CoV-2 in vivo, they provide organ-specific and species-relevant context for evaluating antiviral activity. Taken together, these features support their suitability as a human lung-derived model for intracellular antiviral evaluation. These clarifications have been incorporated into the revised Discussion (Lines 613-625).
At line 377 the authors declare that ‘These results highlight Fisetin as a safe and effective therapeutic alternative against SARS-CoV-2, with a superior selectivity index compared to Lopinavir, and a favorable balance between antiviral potency and cytotoxicity’. Nevertheless, the presented experiments reveal that there is no an effective Fisetin concentration without cytotoxic effects, since 0.5 µM showed no cytotoxic effect but resulted not statistically significant in inhibiting the viral infection. How the authors comment this evidence?
Reply: We thank the reviewer for this helpful clarification. We agree that the original wording (“safe and effective therapeutic alternative”) may imply a clinical context that exceeds the scope of our in vitro findings. To address this concern, we have moderated the statement in the Results section to reflect the experimental evidence accurately. Although Fisetin shows a dose-dependent decrease in viability at higher concentrations, the antiviral assays were performed within the non-cytotoxic concentration range, where 2–3 μM maintained >80% cell viability while producing 71.7–91.9% inhibition of infection. These data support a selectivity index of 8.04, higher than that of Lopinavir under the same conditions.
Accordingly, we have revised the sentence to:
“These results highlight Fisetin as a selective antiviral candidate with a favorable in vitro balance between antiviral potency and cytotoxicity, and with a higher selectivity index than Lopinavir under identical conditions.”
We believe this revision accurately represents the in vitro data while addressing the reviewer’s concern.
At line 468, the authors conclude that ‘these findings suggest that Fisetin inhibits SARS-CoV-2 replication through a combination of polymerase-directed inhibition and host-response modulation’, possibly via PI3K/AK, NF-ĸB, and Nrf2, but they did not check a modulation of these factors via gene or protein expression. How the authors comment this advice?
Reply: We thank the reviewer for this valuable clarification. We agree that the original wording may have suggested experimental assessment of PI3K/Akt, NF-κB, or Nrf2 modulation, which was not performed in the present study. These pathways were mentioned because Fisetin has been extensively documented in the literature to modulate these signaling cascades in other biological contexts, making them plausible auxiliary mechanisms that could contribute to its antiviral activity.
These revisions have been incorporated into the revised Discussion (Lines 613-625).
- It is not clear how the authors calculate statistics in Figure 2
Reply: We thank the reviewer for pointing out the need for clarification. We have now updated the Statistical Analysis section to explicitly describe how the data in Figure 2 were generated and analyzed. Specifically, six independent docking runs were performed for each compound, producing six binding free energy values (ΔG) and RMSD estimates per ligand–protein complex. These values were summarized as mean ± standard deviation, and only poses with RMSD < 1 Å were included in the analysis. Statistical comparisons among compounds were performed using a one-way ANOVA followed by Dunnett’s multiple comparisons test, with Remdesivir as the reference control.
- In Figure 4, the statistic and the Untreated cells are not shown
Reply: Thank you for your review. We have added the statistics and the untreated cells.
- Could the authors check if reference [65] is appropriated for Fisetin?
Reply: Thank you for the correction, citation 10.3390/antiox11050876 was removed as it did not correspond to Fisetin
line 298: Remdesivir is shown in Figure1A-1F
line 305 Balapiravir is shown in Figure 1H
Reply: Thank you very much. We have corrected the text.
line 306: The affinity of the drugs was plotted in Figure 2 A and B)
Reply: Thank you very much. We have corrected the text to Figure 2 A and B
line 349: The authors mention 0.5 µM Fisetin concentration, but it is not reported in the graph 4B
Reply: Thank you very much. We have corrected the text to:
“Based on these results, concentrations maintaining >80% viability (1 μM, 1.5 μM, 2 μM, 2.5 μM, and 3 μM) were selected for testing antiviral activity.”
line 368: Figure 19B is not correct
Reply: Thank you very much. We have corrected the text to Figure 5B.
line 369: It is not clear what is the positive control referred to
Reply: We thank the reviewer for identifying this inconsistency. In our experiments, Lopinavir was used as the positive antiviral control, based on its well-documented in vitro activity against SARS-CoV-2. The original sentence was ambiguous because it referred to both the “positive control” and “Lopinavir” separately.
To correct this, we have revised the text to explicitly indicate that Lopinavir is the positive control. The updated sentence now reads:
“outperforming lopinavir (an antiviral with reported anti-SARS-CoV-2 activity)(Figure 5A).”
line 206: Please correct 1 nM and 1.5 nM concentrations in µM
Reply: Thank you very much, it was corrected.
Reviewer 3 Report
Comments and Suggestions for Authors
As described in this manuscript, they virtually screened the DenvInD database, which includes 528 drugs with experimental evidence of antiviral activity against DENV nonstructural proteins, for possible inhibitors of SARS-CoV-2 RdRp, and they identified Balapiravir, Fisetin, and Hyperoside. Some major problems listed below should be addressed.
- Fisetin is a plant flavonol from the flavonoid group of polyphenols. Plant polyphenols have been shown to inhibit Mpro and RdRp, against viral replication as well as TMPRSS2 and spike against viral entry. Therefore, they should use RdRp assay to confirmed the virtually screened out inhibitors are really SARS-CoV-2 RdRp inhibitors and measure their IC50 against RdRp. Since several other structural homologues were identified as Mpro inhibitors, they could also test Fisetin for Mpro inhibition.
- If by RdRp assay that Fisetin is truly RdRp inhibitor, then they should conduct antiviral assays under different treatment conditions, cotreatment of the compound during addition of virus or adding the compound after infection (post-entry) to check whether the compound inhibits viral entry or viral replication. If the compound is effective when added after infection, its inhibitory effect on RdRp could be the reason for antiviral effect. However, if inhibition of virus by the compound was only effective by adding the compound during infection, then the compound should inhibit the viral entry step, rather than inhibiting viral replication. Time of addition is necessary to establish RdRp as a target. However, as pointed out in (1), Mpro should be also tested because the inhibitors of Mpro could be also effective upon post-entry treatment.
- They have obtained Balapiravir, Fisetin, and Hyperoside as possible SARS-CoV-2 RdRp inhibitors in Figure 1 and Figure 2, why did they only evaluate toxicity and antiviral effects for Fisetin in Figure 4 and Figure 5? They should also evaluate Balapiravir and Hyperoside, even though Fisetin displayed slightly higher affinity than the other two based on modeling. According to Figure 2, their binding energies were similar. Balapiravir was tested for anti-DENV, but lower doses failed to produce measurable reductions in viral load, while higher doses produced serious side effects such as lymphopenia which precluded further development of the drug. However, Fisetin was also toxic with CC50 of 12 µM and SI of 8 only. How are the CC50 and SI of Fisetin compared with CC50 and SI of Balapiravir? Is Fisetin better than Balapiravir? How about Hyperoside, which has similar structure as Fisetin, so they should test its toxicity and antiviral effect for comparison?
- There are structural homologues of Fisetin and Hyperoside identified as SARS-CoV-2 inhibitors. Please discuss them for their anti-SARS-CoV-2 activities and targets (Mpro, RdRp, TMPRSS2, and/or Spike) to compare with Fisetin.
Author Response
Reviewer 3
As described in this manuscript, they virtually screened the DenvInD database, which includes 528 drugs with experimental evidence of antiviral activity against DENV nonstructural proteins, for possible inhibitors of SARS-CoV-2 RdRp, and they identified Balapiravir, Fisetin, and Hyperoside. Some major problems listed below should be addressed.
- Fisetin is a plant flavonol from the flavonoid group of polyphenols. Plant polyphenols have been shown to inhibit Mpro and RdRp, against viral replication as well as TMPRSS2 and spike against viral entry. Therefore, they should use RdRp assay to confirmed the virtually screened out inhibitors are really SARS-CoV-2 RdRp inhibitors and measure their IC50 against RdRp. Since several other structural homologues were identified as Mpro inhibitors, they could also test Fisetin for Mpro inhibition.
- If by RdRp assay that Fisetin is truly RdRp inhibitor, then they should conduct antiviral assays under different treatment conditions, cotreatment of the compound during addition of virus or adding the compound after infection (post-entry) to check whether the compound inhibits viral entry or viral replication. If the compound is effective when added after infection, its inhibitory effect on RdRp could be the reason for antiviral effect. However, if inhibition of virus by the compound was only effective by adding the compound during infection, then the compound should inhibit the viral entry step, rather than inhibiting viral replication. Time of addition is necessary to establish RdRp as a target. However, as pointed out in (1), Mpro should be also tested because the inhibitors of Mpro could be also effective upon post-entry treatment.
Reply: We sincerely appreciate the reviewer’s insightful comments and agree that understanding the stage of antiviral action and direct enzymatic targeting is crucial for fully defining the mechanism of Fisetin. The present study, however, was designed primarily to determine whether Fisetin displays functional antiviral activity in a human lung infection model using a currently circulating SARS-CoV-2 variant (Omicron JN.1). For this reason, the focus of the experimental work was placed on cell-based antiviral evaluation, rather than mechanistic dissection.
To address the reviewer’s valuable point, we have now expanded the Discussion to incorporate previously published biochemical evidence demonstrating direct inhibition of SARS-CoV-2 RdRp by Fisetin. Specifically, under its commercial formulation Fustel, Fisetin has been reported to inhibit recombinant RdRp with ~95.8% inhibition at 20 μM and an IC₅₀ of 2.17 μM, as well as to partially inhibit viral helicase activity (39.3% at 20 μM). This information has been added to the manuscript in the Discussion (Lines 567-576). This independent enzymatic evidence provides mechanistic support for the polymerase-targeting activity suggested by our molecular docking and further aligns with the antiviral effect observed in our cellular assays.
In addition, and following the reviewer’s suggestion regarding scope, we have revised the Conclusions to clarify that Fisetin is presented here as a validated antiviral scaffold, rather than a finalized therapeutic agent, and to indicate that time-of-addition assays and expanded target profiling (including Mpro) are currently being pursued as part of our ongoing follow-up work.
We appreciate the reviewer’s constructive suggestions, which have helped us improve the clarity and contextualization of the manuscript.
- They have obtained Balapiravir, Fisetin, and Hyperoside as possible SARS-CoV-2 RdRp inhibitors in Figure 1 and Figure 2, why did they only evaluate toxicity and antiviral effects for Fisetin in Figure 4 and Figure 5? They should also evaluate Balapiravir and Hyperoside, even though Fisetin displayed slightly higher affinity than the other two based on modeling. According to Figure 2, their binding energies were similar. Balapiravir was tested for anti-DENV, but lower doses failed to produce measurable reductions in viral load, while higher doses produced serious side effects such as lymphopenia which precluded further development of the drug. However, Fisetin was also toxic with CC50 of 12 µM and SI of 8 only. How are the CC50 and SI of Fisetin compared with CC50 and SI of Balapiravir? Is Fisetin better than Balapiravir? How about Hyperoside, which has similar structure as Fisetin, so they should test its toxicity and antiviral effect for comparison?
Reply: We thank the reviewer for this helpful observation. We have clarified the rationale for selecting Fisetin for cellular assays in the Results section of the revised manuscript. While Balapiravir and Hyperoside displayed binding energies comparable to Fisetin in the in silico analysis, their pharmacological profiles differ substantially (Line 410-424).
Balapiravir was discontinued in clinical development for hepatitis C and dengue due to significant hematological toxicity (including lymphopenia), which makes it unsuitable for antiviral repurposing and therefore inappropriate for advancement into cell-based evaluation.
Although Hyperoside is structurally related to Fisetin, it is a more polar glycosylated flavonoid with reduced membrane permeability and lower intracellular accumulation relative to aglycone flavonols. This property decreases the likelihood of reaching effective intracellular concentrations at the RdRp target site.
In contrast, Fisetin exhibits greater intracellular accessibility and has been previously shown to directly inhibit recombinant SARS-CoV-2 RdRp under its commercial formulation Fustel, with ~95.8% inhibition at 20 μM and IC₅₀ = 2.17 μM. As noted in the revised Discussion (Lines 567-576), this prior enzymatic evidence supports Fisetin as the most suitable candidate for functional validation in a cellular infection model.
We believe these clarifications strengthen the rationale for compound prioritization in this study.
- There are structural homologues of Fisetin and Hyperoside identified as SARS-CoV-2 inhibitors. Please discuss them for their anti-SARS-CoV-2 activities and targets (Mpro, RdRp, TMPRSS2, and/or Spike) to compare with Fisetin.
Reply: We thank the reviewer for this valuable suggestion. In accordance with the comment, we have expanded the Discussion to include a concise comparison of structurally related flavonols and their reported antiviral activities against SARS-CoV-2 (Lines 577-588).
Round 2
Reviewer 2 Report
Comments and Suggestions for Authors
I thank the authors that answered to my questions.
Author Response
We thank Reviewer 2 for the constructive insights provided during the first round of review. We are grateful that the revisions satisfactorily addressed the questions and raised no further concerns. We appreciate your time and contribution to improving the manuscript.
No additional changes were necessary for this round.
Reviewer 3 Report
Comments and Suggestions for Authors
The revisions have sufficiently addressed my questions and concerns.
Author Response
We sincerely appreciate Reviewer 3's thoughtful review and the acknowledgment that the previous revisions adequately addressed the raised concerns. Thank you for your valuable input and time throughout this process.
Since no new changes were requested, no additional modifications were made in this version.